# Genetic and Metabolite Variability among Commercial Varieties and Advanced Lines of *Vicia faba* L.

**DOI:** 10.3390/plants12040908

**Published:** 2023-02-17

**Authors:** Eleni Avramidou, Efi Sarri, Ioannis Ganopoulos, Panagiotis Madesis, Leonidas Kougiteas, Evgenia-Anna Papadopoulou, Konstantinos A. Aliferis, Eleni M. Abraham, Eleni Tani

**Affiliations:** 1Department of Forestry and Natural Environment, School of Agriculture, Forestry and Natural Environment, Aristotle University of Thessaloniki, 54124 Thessaloniki, Greece; 2Institute of Applied Bioscience, CERTH, Thermi, 57001 Thessaloniki, Greece; 3Laboratory of Plant Breeding and Biometry, Department of Crop Science, Agricultural University of Athens, Iera Odos 75, 11855 Athens, Greece; 4Institute of Plant Breeding and Genetic Resources, HAO-Dimitra, Thermi, 57001 Thessaloniki, Greece; 5School of Agricultural Sciences, Department of Agriculture Crop Production and Rural Environment, University of Thessaly, 38446 Volos, Greece; 6Laboratory of Pesticide Science, Department of Crop Science, Agricultural University of Athens, 11855 Athens, Greece; 7Department of Plant Science, McGill University, Macdonald Campus, Ste-Anne-de-Bellevue, QC H9X 3V9, Canada

**Keywords:** faba bean, genetic diversity, metabolomics, seed nutritional value, primary metabolites, SCoT molecular markers

## Abstract

*Vicia faba* L. (faba bean) is one of the most promising pulse crops due to its nutritional value and high nitrogen fixation capacity. The aim of the present study was to compare the genetic diversity and the seed metabolite profiles of five genetic materials of faba bean. Specifically, three newly developed advanced lines (KK18, KK14 and KK10) and two commercial cultivars (POLIKARPI and TANAGRA), were evaluated for this purpose. Genetic diversity among populations was assessed by SCoT molecular markers. Through UPGMA dendrogram, genetic distances between populations were estimated. Untargeted metabolomics analysis of the seeds was performed employing GC/EI/MS. The cultivar POLYKARPI exhibited the highest polymorphism. All varieties showed a higher within-cultivars and advanced lines variability than between. POLYKARPI and KK14 had the lowest genetic distances, while KK18 and TANAGRA presented the highest ones. The advanced line KK18 displayed the best nutritional profile, the highest concentration of desirable metabolites (lactic acid and trehalose), the lowest concentration of anti-nutritional factors (oxalic acid) and the lowest concentration of saturated fatty acids (palmitic and stearic acid). According to the results of the present study, KK18 line is a very promising material for further exploration and utilization in breeding programs.

## 1. Introduction

Genetic diversity among species members is a crucial pre-requisite for successful plant breeding. Phenotypic traits and DNA markers, which represent various aspects of genome polymorphism and are occasionally used independently in parental selection, are the two main data sources. However, integration of phenotypic data with genotypic information is highly advised when neutral DNA markers are used for this purpose [1,2]. This aspect becomes even more important when it comes to biochemical characteristics, whose expression is constrained, among others, by the fitness they provide to plants. Combining DNA markers with chemical profiling is thought to be required in such cases, especially for crops with a long history of domestication that include chemical phenotypes commercially relevant [1]. Since noncoding sequences comprise the majority of plant genomes, most changes in molecular marker patterns are predicted to be caused by neutral mutations that are fixed by genetic drift rather than by selection [2].

Faba bean (*V. faba*), also referred to as broad bean, horse bean and field bean, is one of the oldest cultivated plants whose cultivation dates back to 3000 B.C. in Ancient Egypt [3,4,5]. Originating in the Near East, *V. faba* was spread worldwide and is currently cultivated in nearly 70 countries across the world. *V. faba* can be the highest yielding grain legume in the right environment [6,7]. It constitutes a significant resource for agro-ecosystem sustainability, thus providing another advantage over other legumes [8,9,10], due to its ability to adapt to diverse edaphoclimatic conditions [11]. It also has a high capacity for biological nitrogen fixation, i.e., the amount of N fixed by *V. faba* alone was estimated to be comparable to the N amount fixed by the soybean and pea cultivations combined [6,12]. It is one of the edible pulse crops among *Vicia* species, and compared to other legumes, it is distinguished by its high protein content and balanced amino acid profile [13]. The nutritional importance of faba bean is prominent, being a considerably low-cost source of protein. Its seeds contain on average 29% protein as dry matter (DM), lipids, complex carbohydrates, dietary fiber, choline, lecithin, minerals, vitamins and secondary metabolites, including phenolic compounds [14,15], offering a valuable amount of energy [5,16]. On the other hand, *V. faba* is not yet fully exploited as an animal feed due to the presence of some antinutrients, which limit its optimal use [17,18,19]. Current research has focused on the reduction or removal of antinutrients, namely, vicine and convicine [20,21,22], and seed coat tannins [23,24,25], through the development of new cultivars with low amount of antinutritional factors or by using simple processing techniques, such as fermentation, with surprisingly little effort dedicated to improving the protein composition [26].

The exploitation of the genetic diversity of various collections of germplasm has been enhanced by the application of molecular markers [24,27,28]. Although modern molecular markers such as Single Sequence Repeat (SSR) and Single Nucleotide Polymorphism (SNP) have been developed, the large genome of the genus *Vicia* (13Gb) is a barrier to the detailed study of genetic diversity at genome level [24,27,28]. Yang et al. [28], used 94 Expression Sequence Tag-Single Sequence Repeats (EST-SSR) to classify 32 *V. faba* genotypes based on their genetic distances and revealed clusters according to their geographical origin [29]. Furthermore, 657 SNPs utilized in 45 accessions of *V. faba*, also reported groups based on their geographical origin, with Mediterranean Basin and Near Eastern genotypes grouping together apart from Chinese accessions [30]. In addition, Wang et al. [31], studied *V. faba* landraces and identified a strong linkage between genetic diversity and the geographical origin of the accessions being studied. However, the population structure of Mediterranean *V. faba* landraces as well as their distance from wild related species is poorly analyzed.

The genetic diversity of *V. faba* has been widely studied for characteristics such as the flower morphology, seed size, resistance to stress conditions as well as the content of nutrients [32]. A variety of molecular markers (RAPD, SSR, EST-SSR, TRAP, IRAP and AFLP) have been used to study the genetic diversity of *V. faba* genetic material [33,34,35,36,37]. Tomas et al. [36], applied 12 TRAP primer combinations amplified and revealed a high level of polymorphism among the studied genotypes. Moreover, Inter-simple sequence repeat (ISSR) markers were also utilized to explore genetic diversity of *V. faba* accessions [31,38,39,40]. Start codon targeted (SCoT) polymorphism is a new molecular marker, and it was developed based on the short-conserved region flanking the adenine–thymine–guanine (ATG) start codon in plant genes [41,42]. SCoT markers are a valid choice for the analysis of the genetic diversity of *V. faba* species, distinguishing and highlighting various genotypes that can be used in breeding programs aimed at meeting the needs of producers [43]. SCoT resembles the random amplified polymorphic DNA (RAPD) and inter simple sequence repeat (ISSR) since a single primer is used as forward and reverse [44]. SCoT markers have been used for genetic diversity studies in other *Vicia* species, such as *Vicia sativa* [43]. Regarding *V. faba*, no studies have been published on the identification of SCoT markers related to metabolite traits.

Recently, there has been a growing interest in functional foods, vegetables and legumes, due to their valuable protein composition and richness in metabolites. Moreover, seed quality and metabolism are of fundamental importance for agriculture [45]. Seed metabolomics analysis has undoubtedly become an indispensable tool in identifying biomarkers that are useful in seed quality improvement and in combating malnutrition [46,47,48,49,50,51,52,53,54,55]. Furthermore, by correlating with molecular markers and agronomical traits, metabolites can also be employed to offer comprehensive information (as selection criteria in pre-breeding procedures) [56] regarding the potential use of the genetic material. The possibility to perform statistical analysis on large-scale datasets using correlative approaches, has facilitated the identification of metabolites linked to specific genotypes [57].

Legumes are considered an excellent source of nutrients and bio active primary and secondary metabolites belonging to various classes, i.e., amino acids, fatty acids, organic acids, phenolic compounds, etc. [58]. Additionally, non-GM legume crops, which are locally produced and can be protein alternatives and nutrient-rich sources, are in great demand for the improvement of the self-sufficiency and sustainability of farming systems [59,60,61,62]. Recent metabolomics studies have focused on the phytochemical/antioxidant properties of different *V. faba* seed germplasm [5,63] and highlighted the importance of increasing its consumption by humans. However, to the best of our knowledge, there are no such studies regarding *V. faba* seed metabolomics and its impact on supplements for animal feed. The only studies referring to *V. faba* used for animal feed, are mainly focused on ensiled products and on specific metabolites [59,64], or on its use as a protein alternative to soybean meals [60].

The aim of the present study was to identify and characterize the genomic and metabolic diversity in selected populations (both advanced lines and commercial varieties) of *V. faba* and to unravel any possible relationships between the two sets of data. Moreover, we investigated the usefulness of metabolomics regarding the identification of material that can be further exploited into breeding programs, by setting as main selection criteria the low saturated fatty acids content and other non-nutritive factors (i.e., low oxalic acid and myo-inositol content) for further use for human consumption and animal feed.

## 2. Results

### 2.1. Genetic Diversity Estimated by SCoT Markers

A total of 103 loci were obtained from the 6 selected SCoT primers. Of all loci analyzed, 58.25% were polymorphic and 41.75% were monomorphic within or between the commercial cultivars and the advanced lines. Gene diversity (GD) within the studied genetic material of *V. faba* ranged from 0.205 to 0.269 and Shannon index (I) from 0.300 to 0.392 (Table 1). The cultivar POLIKARPI exhibited the highest level of polymorphism (P = 66.02%) and TANAGRA the lowest (P = 50.49%). In addition, POLIKARPI showed slightly higher levels of genetic variation for the Shannon’s information index (0.392) compared to TANAGRA and to advanced lines (Table 1).

According to the AMOVA analysis, 69% of the total genetic variation was attributed to differences within the cultivars and the advanced lines, and the rest (31%) was attributed to differences among them (Table 2).

Genetic differentiation (Nei’s genetic distance) [65] ranged from 0.006 (POLIKARPI to KK14) to 0.389 (KK18 to TANAGRA) (Table 3).

The UPGMA dendrogram, based on the genetic distances among the studied genetic materials, depicted three main clusters (Figure 1). The first cluster included cultivar POLIKARPI (K3), the second one, line KK18 (K1) and the third one KK14 (K4) line, which was also divided into two main sub clusters in which were KK10 (K5) line and the cultivar TANAGRA (K2). According to the PCoA, PC1 and PC2 accounted for 13.60% and 26.84%, respectively, of the total variation. The studied genetic material formed distinct clusters (Figure 2) which is in accordance with the UPGMA dendrogram.

STRUCTURE analysis with SCoT markers was performed without prior information of samples, and the highest likelihood of the data was obtained for K = 4. After K = 4, the results are not consistent between different runs. Evanno test based on ΔK (Figure 3) revealed that the optimum number of K is 4. Y axis represents DeltaK = mean[IL″(K)]/sd[L(K)] and X cluster the number of clusters.

The data set was partitioned into clusters: individuals from TANAGRA and KK10 belong to the same cluster, whereas KK18, KK14 and POLIKARPI C2, C3, K1 and K2 tended to be classified in separate clusters. Low levels of admixture were observed in KK10 and KK14 and almost no admixtures in TANAGRA. On the other hand, a moderate level of admixture was detected in POLIKARPI and KK18 (Figure 3).

### 2.2. Correlation Analysis

Mantel’s test was used to evaluate the correlation between the markers (genomic and meta-bolomic) (r-statistic). Metabolic profiles and matrices derived from SCoT data showed a low correlation between the genetic distances and the metabolites (R = 0.122, *p* < 0.05, 99 iterations).

### 2.3. Metabolomics Analysis

In total, 130 metabolite features of faba bean seeds were reproducibly recorded and subjected to multivariate analysis for the discovery of trends and the corresponding biomarkers. A set of representative metabolite profiles of *Vicia faba* L. (PMG-01-23) in “*.cdf” format, can be accessed from the repository of the Pesticide Metabolomics Group (https://www.aua.gr/pesticide-metabolomicsgroup/Resources/default.html). Metabolomics analysis revealed a strong discrimination among the metabolite profiles of the analyzed genetic material lines’ seeds (Figure 4). The tight grouping that was observed is indicative of the performance of the applied bioanalytical protocols. Results of OPLS-DA were confirmed by the corresponding HCA dendrogram (Figure 5). Two distinct groups were obtained. The first one includes the advanced lines KK18 and KK10 and it is linked to cultivar TANAGRA. The second one includes the cultivar POLIKARPI and the advanced line KK14. Furthermore, the metabolites-biomarkers of the analyzed genetic material were discovered based on the VIP values (Figure 6).

Ten metabolites exhibited the highest variability (Table 4) and were classified into two main chemical groups, organic acids and sugars. It was noteworthy that the lowest concentrations of saturated fatty acids such as palmitic and stearic acids were detected in KK18 line, whereas KK10 accumulated the highest amounts. Moreover, both KK18 and TANAGRA had the lowest content in oxalic acid, whereas KK14 the highest. Finally, KK18 presented medium to low concentrations of myo-inositol and D-myo-inositol phosphate respectively, with TANAGRA having the lowest content in myo-inositol.

## 3. Discussion

The development of *V. faba* germplasm with high nutritional value is of high importance for future applications in human and animal welfare. Solid knowledge on the germplasm resources used in breeding programs is a prerequisite for their effective management. To the best of our knowledge, this is a first report on the genetic diversity and composition of *V. faba* cultivars and advanced breeding lines using SCoT markers combined with metabolomics analysis of their seeds.

Several dominant markers have been used to assess genetic diversity of *V. faba* germplasm. According to the results of the present study, 58.25% of the alleles were polymorphic and this percentage was lower or comparable to that of SRAP markers (97.2) [66], TRAP markers (55.2%) [35], RAPD markers (75.96%) [67], S-SAP markers (71.10%) [68] and ISSR markers (85.5%) [40]. Furthermore, Terzopoulos and Bebeli [38] studied the genetic diversity of Greek local populations of *V. faba* by using ISSR markers and according to their results polymorphism was 98.9%. The Shannon diversity index based on SCoT analysis was, on average, 0.35 in the present study, while based on TRAPs was 0.15 [35], on RAPDs was 0.29 [67], on S-SAPs was 0.207 [68] and on ISSRs was 0.41 [40]. The differences recorded in the assessment of genetic diversity can be attributed to marker specificity effect on the manner of polymorphism [69] and/or to the diversity of germplasm that was used.

According to the AMOVA, higher genetic diversity was recorded within (69%) than among (31%) the studied populations. Similarly, a higher level of genetic diversity within than among populations of *V. faba* have been reported by Terzopoulos and Bebeli [38] for Greek local populations with ISSRs, by Bachouchi et al. [67] for Tunisian populations with RAPDs, by Gresta et al. [70] for Sicilian landraces with AFLPs and by Ouji et al. [68] for Tunisian populations with S-SAPs, which is congruent with our results. *V. faba* is a partially allogamous species with a high and variable amount of outcrossing [71], which is depending on genetic and environmental factors [35]. The results of the present study are suggesting that the studied populations were predominantly allogamous as higher genetic diversity within than among population was expected for outcrossing species [72].

According to UPGMA cluster analysis and PCoA the studied genetic materials were genetically distinguishable especially the cultivar POLIKARPI and the advanced line KK18. On the other hand, cultivar TANAGRA and the advanced line KK10 presented the highest similarity. STRUCTURE results further verified UPGMA and PCoA results.

In this study, we identified and analyzed ten main metabolites, discerned into two main categories (organic acids and sugars). Interestingly, nine of them showed statistically significant differences between the five varieties. As previously mentioned, due to lack of studies on *V. faba* seed profile of primary metabolites, we are comparing our findings with studies on seeds’ metabolite composition of other grain legumes.

Organic acids comprise a crucial nutritional factor that seems to be related to many metabolic pathways and physiological functions in plants. Five main acids (L-lactic acid, gluconic acid, malic acid, palmitic acid, stearic acid and oxalic acid) were mainly recorded in the present study. In a very recent review of Emkani et al. [73], it is well understood the importance of lactic acid for the quality of the final legume product regarding organoleptic and biochemical traits. In the current study the KK18 line seems to be the one with the highest accumulation of lactic acid.

Saturated fatty acids are well studied (palmitic and stearic acid) since their implication in human health is well documented. Palmitic acid seems to be the most abundant among bean varieties [74] and *Vicia* species [75]. Many researchers have attempted to create varieties low in saturated fatty acids through modern breeding methods [76,77]. KK18 was found to have the lowest levels of saturated fatty acids (palmitic, stearic) and as a result, it provides the possibility of having the greatest proportion of essential beneficial unsaturated fatty acids (i.e., oleic, linolenic) [78].

Oxalates, such as oxalic acid, are referred to as antinutritional factors causing hypocalcemia, mineral bioavailability disorders and stone disorders [79,80,81]. In our study, the advanced line KK14 had the highest concentration of oxalic acid, as well as other antinutritional factors such as total phenols and tannins [82]; thus, it is inappropriate for further use in breeding programs.

Sugars are basic components of legumes seeds and are important quality factors for food and feed [83]. Through metabolome analysis, we identified significant differences between the populations for two sugars, myo-inositol and D-mannitol. According to the research of Ali et al. [84] lower levels of myo-inositol, in transgenic lines of rice, are related to lower levels of phytic acid and higher accumulation of minerals. A very recent study suggests that varieties with low concentrations of phytic acid are strongly recommended for animal feed [85]. In our work, variety TANAGRA showed the lowest concentration of myo-inositol; on the contrary, line KK10 showed the highest. D-mannitol is a sugar alcohol that seems to have an impact on animal feed digestibility [86,87]. TANAGRA exhibited the highest level of D-mannitol and POLIKARPI the lowest.

Association analysis was implemented through a Mantel test between genetic distances of the studied germplasm and metabolite data. Results demonstrated small correlation (R = 0.122). In several studies, no correlation was found between genetic analysis and metabolomic profile [1,2,56]. This can be attributed to the fact that markers, such as ScoT markers, cannot unravel differences in gene function that can cause variations in metabolic profiles. Thus, the latest studies use molecular markers that unravel polymorphic alleles linked to specific secondary and primary metabolites responsible for important agronomic traits such as heat stress tolerance [88]. Moreover, a recent review tried to link SNPs with metabolic traits by integration of genetic variants in genome-scale metabolic models [89]. The identification of markers related to specific metabolites of great agronomic importance (i.e., high unsaturated fatty acid content) would provide a useful tool for breeding programs using Marker Assisted Selection.

Additionally, the increased efficiency of metabolomic tools will facilitate future analysis and will provide more insight into the nature of metabolite networks influenced by the genetic diversity underlying seed quality. As previously discussed, line KK18 harbors high intra-population genetic diversity possibly impacting on the chemical composition of desirable metabolites. These metabolites can be used as metabolic markers in modern breeding strategies, i.e., as selection indices in advanced generations of breeding for seed quality.

## 4. Materials and Methods

### 4.1. DNA Isolation and Markers Analysis

One gram of fresh young leaves of five *V. faba* populations (five individuals per population), three advanced lines developed by the company AGROLAND SA (namely KK18, KK14, KK10) and two commercial varieties (POLIKARPI and TANAGRA) developed by the Institute of Industrial and Forage Crops (IIFC, belongs to the Hellenic Agricultural Organization “Dimitra”) were grinded with liquid nitrogen and saved in −20 °C. The procedure described by Doyle and Doyle [90] was used for the isolation of total genomic DNA. The amount of DNA was quantified by a UV-Vis Spectrophotometer (Q 5000) (Thermo Fisher Scientific, San Jose, CA, USA) and then the samples were diluted to a 20 ng/μL working concentration.

PCR for SCoT analysis was performed in a total volume of 20 μL containing 20 ng total genomic DNA, 100 mM of each dNTP, 1.5 mM Mg, 10 μΜ of primers and 5 u/μL Kapa taq (Kapa Biosystems Pty (Ltd), Cape Town, South Africa). To study intra- and inter-population diversity, 6 SCoT primers (SCoT14, SCoT15, SCoT61, SCoT66, SCoT30 and ScoT33) were used for PCR amplification. According to the literature review, the specific SCoT markers were selected due to their high polymorphic for legume species and to their use in a wide range of plant species. PCR amplifications were performed in a SureCycler 8800 (Agilent Technologies Inc., Santa Clara, CA, USA) as follows: initial denaturation of 5 min at 94 °C followed by 40 cycles of 30 s at 94 °C, annealing at 50 °C for 90 s, and extension at 72 °C for 90 s. A 5 min step at 72 °C was programmed as a final extension.

Amplification products were separated by electrophoresis on 1.5% agarose gel and stained with ethidium bromide. Gel images were placed in a UVItec Transilluminator and a 100 bp or 1 Kb DNA ladder (Invitrogen, Carlsbad, CA, USA) was used as a size marker.

Binary data points denote the presence/absence of each distinguishable band across all samples for the same primer, in both replicate sets of amplifications.

### 4.2. Data Analysis

The percentage of polymorphic loci (P), effective numbers of alleles (NE), gene diversity (expected heterozygosity, HE), Shannon’s diversity index (I) and unbiased genetic distances according to Nei [65] were calculated using GenALEx ver.6.51b2 [91]. The hierarchical distribution of genetic variation among and within populations was also characterized by analysis of molecular variance (AMOVA) [92,93] using the GenALEx ver. 6.51b2 software [91], with variation being examined among and within populations. The tests were implemented using estimates of ΦST, based on distances calculated from allelic data. Tests of significance were performed using 999 permutations within the total dataset. A cluster analysis using an unweighted pair-group method with arithmetic averaging (UPGMA) [94] was carried out using the software GenAlex ver. 6.51b2. Dendrograms were generated using Mega ver11 [95]. The genetic structure of *V. faba* populations was analyzed by performing principal coordinate analysis (PCoA) using GenAlEx ver.6.51b2 [91] based on standardized covariance of genetic distance for dominant markers.

A Mantel test [96] was conducted for the comparison of genetic distances with metabolites using the PopTools version 3.2.5 [97] with 99 iterations.

The number of genetically homogeneous group (K) was inferred using the program STRUCTURE v2.3.3 [98]. This program was run using the admixture model, and 10 repetitions of 1,000,000 iterations afterwards following a burn-in period of 500,000 iterations. Other parameters were set to default values. K was determined using the ad hoc statistic DK as in Evanno et al. [99]. Results analyzed with CLUMPAK [100] and visualized in pophelper [101].

### 4.3. Plant Material and Experimental Conditions for Metabolomic Analysis

Five *V. faba* populations were evaluated in randomized complete blocks design (RCBD) with four replications in three locations (South Greece: Athens, Central Greece: Larisa, North Greece: Thessaloniki) for two consecutive years (November–June 2018/2019 and November–June 2019/2020). Field experimental conditions and design were presented in detail by Papastylianou et al. [102]. Seeds of the second growing season from the three locations were used for metabolomic analysis. The seeds were harvested manually when each variety reached its maturity stage [102]. Seeds from five plants from each location were bulked for each population.

### 4.4. Extraction of Seeds’ Metabolites and Sample Preparation for GC/EI/MS Metabolomics Analysis

*V. faba* seed metabolites were extracted and prepared for GC/EI/MS metabolomics following previously described protocols [103,104]. Briefly, seeds were pulverized to a fine powder in a blender and then, a portion (150 mg) was transferred to plastic Eppendorf tubes (2 mL). In total, six samples were prepared for each population and one quality control sample. Briefly, a mixture of methanol-ethyl acetate (500 μL, 50:50, *v*/*v*) (Carlo Erba Reagents, Val de Reuil, France) was used for metabolite extraction. The extracts were sonicated in an ultrasonic bath (Branson 1210, Danbury, CT, USA) for 20 min followed by 2 h of stirring (150 rpm, at 24 °C) in a horizontal rotary shaker (GFL 3006, Gesellschaft für Labortechnik mbH, Burfwedel, Germany). Filtering followed (0.2 μm pore diameter, Macherey-Nagel, Duren, Germany) and the resulting extracts were evaporated to dryness using a refrigerated vacuum concentrator (Labconco, Kansas City, MO, USA). The dry extracts were derivatized following a two-step process [103,104] using methoxylamine hydrochloride (98.0%, *w*/*w*) [20 mg mL^−1^ in pyridine (98.0%, *w*/*w*)] (Sigma-Aldrich Ltd., Steinheim, Germany) and N-Trimethylsilyl-N-methyl trifluoroacetamide (MSTFA). The samples were transferred to glass microinserters (200 μL, Macherey-Nagel) into glass autosampler vials (2 mL) for analyses. Additionally, selected analytical standards of primary plant metabolites were used for the absolute annotation of metabolite features.

### 4.5. GC/EI/MS Metabolomics Analysis of Seed Extracts

Metabolomics analysis was performed using an Agilent 6890 analyzer (Agilent Technologies Inc., Santa Clara, CA, USA) applying previously described settings). Samples (1 μL) were injected on column at a 5:1 split ratio. The temperature of the oven was 70 °C, stable for 5 min, increased to 295 °C at a 5 °C min^−1^ rate, stable for 5 min. Full scan mass spectra were acquired over the range 50–800 Da (4 scans s^−1^).

The obtained GC/EI/MS total ion chromatograms (TIC) were deconvoluted using the software AMDIS v.2.66 and the metabolite libraries of the National Institute of Standards and Technology library, NIST ‘08 (NIST, Gaithersburg, MD, USA). Their processing for metabolomics analysis was performed using the MS DIAL v.4.80 software [105]. The detection of trends and the discovery of biomarkers of the analyzed populations were performed by multivariate analyses using the bioinformatics software Simca v.17.0 (Sartorius AG, Goettingen, Germany) according to previously presented pipeline [103,104].

### 4.6. Statistical Analysis

The quality traits data were checked for regularity with a Shapiro–Wilk test, homogeneity with Levene and Barlett tests and were found to meet both criteria, as well as for their independence which is ensured through the Random Complete Group design that we implemented in the experiment. This was followed by ANOVA test and comparison of the mean values of the characteristics of the varieties with the method of Minimum Significant Difference (LSD). The statistical processing of the data was performed through the JMP program.

## 5. Conclusions

Two commercial cultivars and three advanced breeding lines were evaluated, comparing their genetic diversity and their main metabolites.

SCoT markers are useful tools for the detection of genetic diversity among faba bean germplasm and help in studying genetic relationships. Genetic diversity is essential for the creation of new varieties with interesting characteristics. The commercial cultivars POLIKARPI and TANAGRA exhibited higher genetic diversity compared to advanced lines. Genetic and metabolomic analyses give similar results regarding the relationships between the studied genetic material. TANAGRA and KK10 are the genetic material with high similarities both at genetic and metabolic levels. Moreover, POLIKARPI and KK18 are genetically and metabolically the most distinct from the others.

On the other hand, line KK18 showed the highest accumulation of desirable nutritional compounds such as lactic-acid and trehalose and the lowest for anti-nutritional factors (oxalic acid). Additionally, KK18 is characterized by high seed yield under low rainfall environments.

This study showed that KK18 is the most promising line for further exploitation due to the production of highly nutritional quality seeds and high seed yield under low rainfall environments [102]. On the other hand, both the cultivars POLIKARPI and TANAGRA can be further used in breeding programs due to their high genetic diversity. Furthermore, TANAGRA is a stable cultivar in variable environments with high seed yield productivity [102]. There is no particular connection between the studied genotypes. Thus, KK18 that has genetic distance with the commercial varieties exhibits many interesting characteristics that can be exploited in a breeding program of crosses to widen the genetic base on which the targeted selection can take place. The present study is a first step towards the development of an efficient breeding program combining genetic variation studies and metabolomic profile in faba bean for use in human nutrition and animal feed.

## Figures and Tables

**Figure 1 plants-12-00908-f001:**
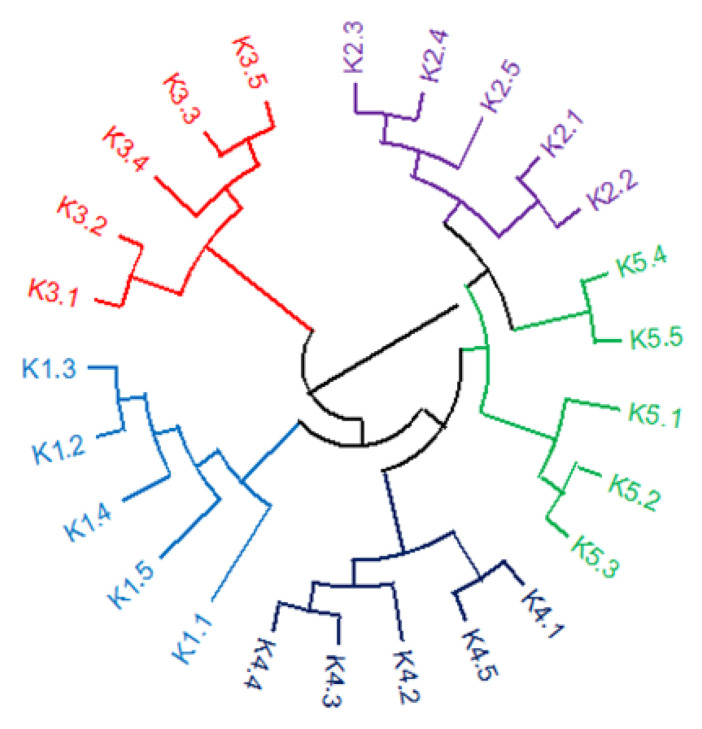
UPGMA dendrogram based on Nei’s genetic distance [65] of *V. faba* populations of advanced lines (K1: KK18, K4: KK14, K5: KK10) and commercial cultivars (K2: TANAGRA, K3: POLIKARPI).

**Figure 2 plants-12-00908-f002:**
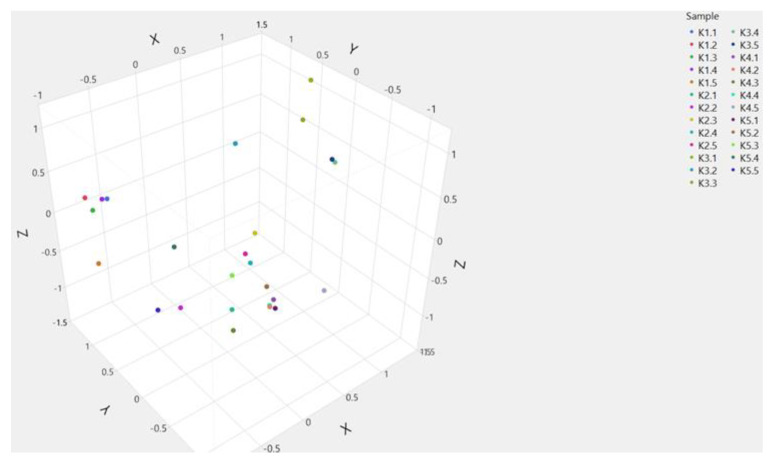
Principal coordinate analysis (PCoA) of *V. faba* cultivars and advanced lines using 6 SCoT markers. Advanced lines (K1: KK18, K4: KK14, K5: KK10) and commercial cultivars (K2: TANAGRA, K3: POLIKARPI).

**Figure 3 plants-12-00908-f003:**
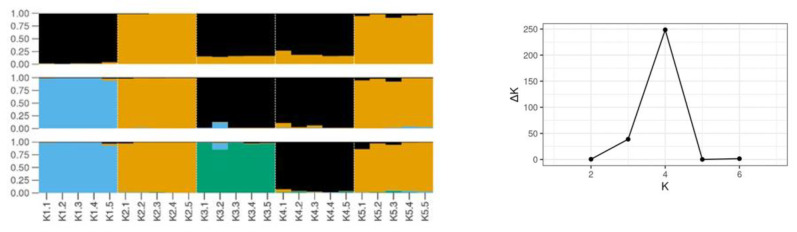
STRUCTURE plots for K = 2, K = 3 and K = 4 for SCoT molecular and five genetic materials of *V. faba*. Each vertical bar represents a single individual (n = 25) and the probability of its membership in four clusters. Advanced lines (K1: KK18, K4: KK14, K5: KK10) and commercial cultivars (K2: TANAGRA, K3: POLIKARPI). DeltaK = mean [IL″(K)]/sd[L(K)]. DK values for different K calculated using the Evanno method. Number of clusters (K).

**Figure 4 plants-12-00908-f004:**
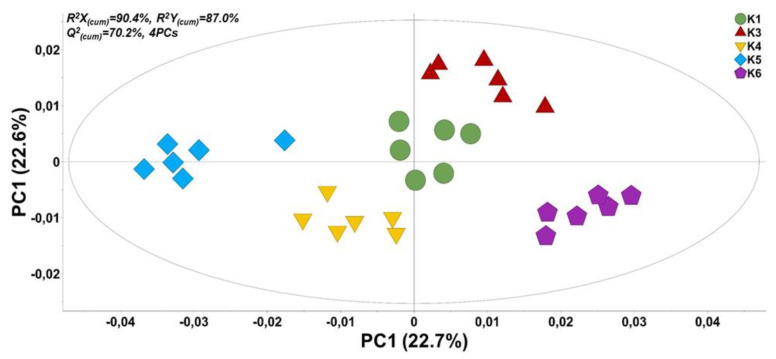
Orthogonal partial least squares discriminant analysis (OPLS-DA). PC1/PC2 score plot displaying the differences among the GC/EI/MS-recorded metabolite profiles of the analyzed *Vicia faba* seeds of five cultivars and advanced lines (K1: KK18, K3: KK10, K4: KK14, K5: POLIKARPI, K6: TANAGRA) at 95% confidence interval. In total, six pooled samples and one quality control sample were analyzed per line. The ellipse represents the Hotelling’s T^2^ [PC(s); principal component(s), Q^2^(cum); cumulative fraction of the total X’s variation that can be predicted, R^2^X and R^2^Y; fraction of the sum of squares of X’s and Y’s explained by the components].

**Figure 5 plants-12-00908-f005:**
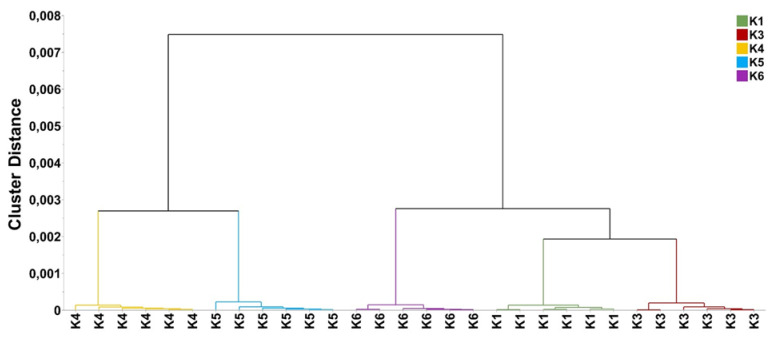
Hierarchical cluster analysis (HCA) dendrogram for the GC/EI/MS-recorded metabolite profiles the analyzed *Vicia faba* seeds of the five cultivars and advanced lines (K1: KK18, K3: KK10, K4: KK14, K5: POLIKARPI, K6: TANAGRA). The Ward’s linkage method was applied. In total, six pooled samples and one quality control sample were analyzed per line.

**Figure 6 plants-12-00908-f006:**
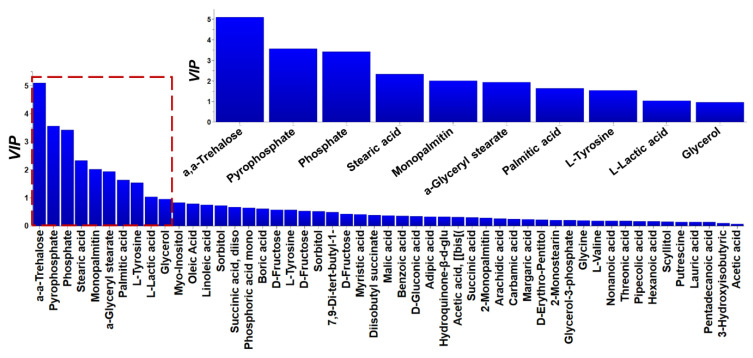
Annotated *Vicia faba* metabolites exhibiting the highest leverage on the observed discrimination among the analyzed lines’ seeds based on the variable influence on projection (VIP) plot. Metabolites are displayed according to their VIP values in descending order. Confidence intervals have been derived from jack-knifing (P < 95%). Magnification of the red dashed area is displayed in the upper plot.

**Table 1 plants-12-00908-t001:** Genetic diversity of cultivars and advanced lines of *V. faba*.

Populations	N ^(1)^	NPB	No. Private Bands	P (%)	Shannon Index (I)	GD
ΚΚ18	5	88	5	51.46	0.319	0.222
TANAGRA	5	78	0	50.49	0.300	0.205
POLIKARPI	5	81	4	66.02	0.392	0.269
ΚΚ14	5	75	0	62.14	0.373	0.256
ΚΚ10	5	83	1	61.17	0.370	0.255
Mean	5	81		58.25	0.351	0.241
Species level	25					

^(1)^ N: number of individuals; NPB: Number of polymorphic bands; P (%): Percentage of polymorphic bands; GD: gene diversity; I: Shannon’s information index.

**Table 2 plants-12-00908-t002:** Analysis of molecular variance results for the studied genetic material of *V. faba* based on SCoT markers.

Source	Df ^(1)^	SS ^(2)^	MS ^(3)^	Est. Var.	%
Among Pops	4	202.080	50.520	6.996	31%
Within Pops	20	310.800	15.540	15.540	69%
Total	24	512.880		22.536	100%
Stat	Value	*p* (rand >= data)			
PhiPT	0.310	0.001			

^(1)^ Df: degree of freedom; ^(2)^ SS: sum of squares; ^(3)^ MS: mean of squares.

**Table 3 plants-12-00908-t003:** Estimates of pairwise Nei’s genetic distance (below the diagonal) and FST values (above the diagonal) within overall groups of *V. faba* species.

KK18	TANAGRA	POLIKARPI	KK14	KK10	
0.000					KK18
0.389	0.000				TANAGRA
0.357	0.355	0.000			POLIKARPI
0.344	0.322	0.265	0.000		KK14
0.321	0.169	0.301	0.244	0.000	KK10

**Table 4 plants-12-00908-t004:** Statistically significant differences in main metabolites among five *V. faba* populations. Six replications were performed per population and the different letters designate statically significant differences performing the LSD test (a = 0.05).

Metabolites	Relative Concentration (×10^−3^ mol/L)				
	KK18	KK10	KK14	POLIKARPI	TANAGRA
a-a-trehalose	771.08 ^a^	719.19 ^b^	677.44 ^c^	758.64 ^a^	721.56 ^b^
Palmitic acid	24.98 ^c^	29.51 ^ab^	30.25 ^a^	28.32 ^ab^	28.64 ^ab^
Oxalic acid	18.51 ^b^	24.79 ^ab^	28.57 ^a^	25.66 ^ab^	18.99 ^b^
D-myo-inositol phosphate	15.21 ^b^	17.80 ^ab^	16.80 ^ab^	16.06 ^ab^	18.83 ^a^
Stearic acid	12.08 ^c^	16.41 ^a^	14.22 ^b^	15.07 ^ab^	13.50 ^bc^
L-lactic acid	5.42 ^a^	3.77 ^bc^	3.20 ^c^	2.98 ^c^	4.75 ^ab^
Myo-inositol	4.06 ^b^	4.72 ^a^	3.46 ^bc^	3.81 ^b^	2.95 ^c^
D-Gluconic acid	3.76 ^b^	5.18 ^a^	3.45 ^bc^	2.44 ^c^	3.44 ^bc^
Malic acid	1.93 ^c^	1.32 ^c^	3.11 ^b^	1.53 ^c^	4.48 ^a^
D-Mannitol	1.91 ^b^	3.17 ^a^	2.86 ^a^	1.52 ^b^	3.25 ^a^

## Data Availability

The datasets generated during and/or analyzed during the current study are available from the corresponding author on reasonable request.

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
