# Peer review of "Genetic and Metabolite Variability among Commercial Varieties and Advanced Lines of Vicia faba L."

_plants, 2023, doi:10.3390/plants12040908_

Round 1

Reviewer 1 Report

This manuscript describes the results of an intense genetic study of the genetic structure of five populations of Vicia faba. Genetic relations and overall manifestations of variability were explored using SCoT markers along with an analysis of the nutritional value of seeds from these populations. However, the rationale for the selection of these genetic entities is never explained in the manuscript. The populations were obtained from commercial sources (breeding lines) and from government organizations. The consequences of the interrelationship studies are questionable since the reader has no idea how the populations were derived or why they were selected in the first place. Otherwise, this is an interesting and impactful study of the partitioning of genetic variability and seed nutritional value among a very small panel of interrelated populations of V. faba.

Abstract

Lines 19-20) “A first step was done to compare the genetic diversity and the seed metabolomics profil of five genetic materials of faba bean.” Was there a second step? It appears that the project was undertaken to compare genetic diversity and seed metabolomics parameters among five populations of V. faba. The rationale for the choice of genotypes included is not provided. This is important for readers to understand the scope of the project. Also, “profil” (sic) is misspelled.

Lines 22-24) “Through this research, we were able to use genomic and metabolomic methods to characterize those Vicia faba genetic materials. The most promising genotypes, capable of creating future breeding programs for very valuable traits, can be chosen using these techniques.” Delete.

Lines 27- 28) “All the studied genetic materials showed high genetic variability.” Delete due to redundancy with subsequent sentences.

Introduction

Paragraph 1) The breadth and depth of precepts provided in this paragraph are not adequately documented by the citation provided; more documentation in published literature is needed.

Paragraph 2) No new paragraph is needed. The reviewer suggests that “composed” be changed to “comprised”.

Line 53) “Fava bean…” the common name is given as “faba bean” in the Abstract.

Line 62 and elsewhere in the manuscript) “…Vicia…” should be italicized. Also, V. faba is not italicized in many places in the manuscript.

Line 65) “…on a…” change to “…as…”.

Line 74) “…genetic material…” change to “…germplasm…”.

Line 77) “…species Vicia…” change to “…genus Vicia…”.

Line 97) “…SCoT markers are a valid choice…”. The reviewer agrees with the authors on this assertion but the reasons given do not include that they represent the coding region of the genome.

Line 107) “…of fundamental importance for…”. For what?

Line 111) “…condensed information…” an explanation of this statement is needed.

Line 117) “…Popoola et al.” Delete.

Results

Table 1) The formatting of the table legend is not correct.

Tables 1 and 2) Table 1 refers to the experimental factor as “Genetic Materials” whereas Table 2 refers to this factor as “Populations”.  The terminology should be internally consistent.

Table 3) If pairwise estimates of genetic distances between the populations is reported below the diagonal (null comparisons), what are the values reported above the diagonal? Why are they different than the pairwise values above the diagonal?

Lines180-181) “Thus, we conclude that K= 4, that is also what the Evanno analysis gives (Figures 3 and 4).” This sentence is confusing and should be rewritten to be clearer.

Discussion

Lines 257-258) “It is of outmost importance a breeding program towards the development of V. faba varieties, with high nutritional value for human consumption and animal feed.” Change to “The development of V. faba germplasm with high nutritional value is of high potential importance for future human and animal welfare.”

Line 261) “…structure of V. faba germplasm ..”. Change to “…composition of V. faba cultivars and advanced breeding lines…”.

Lines 261-262) “…(3 advanced lines and 2 commercial cultivars)…” Delete.

Line 307) “…Bai et al.” Delete.

Line 308) “Oxalic acid is a type of oxalate, oxalates are referred as antinutritional factors causing …” Change to “Oxalates such as oxalic acid cause…”.

Line 314) “On the contrary, KK18 had the lowest concentration.” This sentence seems to be extraneous and should be deleted.

Lines 327-328) “…neutral markers, such as ScoT markers…” Is it correct to refer to ScoT markers as “neutral” since they emanate from the control regions of expressed sequences?

Line 340) “…high inter-genetic diversity…” the reviewer believes it is more accurate to state “…high intra-population genetic diversity…”.

Materials and Methods

Line 348) “…Institute of Industrial and Forage Crops (IIFC)…”; referring to the government of Greece?

Line 353) “…SCOT…” change to “…SCoT…”.

Conclusions

Line 459) “It emerges from this study, that…” change to “This study showed that…”.

Author Response

We thank very much the reviewer for the fruitful comments. Below we attach the responses to all the comments. 

Sincerely

The corresponding author

Eleni Tani

Reviewer 2 Report

The manuscript presentic the genetic distance and the diffrent composition of 5 fava varieties/lines is globally well written, but some sentences are too general and not supported with the litterature nor the results giving to false conclusions both in the introduction and in the conclusions. These parts should be rewritten according to the comments in the attached documents.

In the discussion it will be interesting to find more comparison of the studied varieties to the litterature in terms of composition, to see the relative level of diversity that exists and not only based on 5 "accessions".

Some minor details have to be completed in the material and methods section as well as in the results (form of tables and figures).

Author Response

We thank the reviewer very much for the fruitful comment.

Below we attach the answers to all of the comments.

Sincerely

The corresponding author

Dr Eleni Tani

Round 2

Reviewer 2 Report

Only one form correction:

Line 363, it is not necessary to provide the complete adress, only the constructor name, the city and the country